# OpenReview forum: "Can AI Perceive Physical Danger and Intervene?"
_ICLR.cc/2026/Conference — ICLR 2026 Conference Desk Rejected Submission_

### Official Review · Reviewer_oYkf · 2025-10-30

**Soundness:** 2
**Presentation:** 2
**Contribution:** 1
**Rating:** 0
**Confidence:** 5

**Summary:**

The authors propose ASIMOV-2.0, a new benchmark dataset aimed at evaluating the physical safety awareness of foundation models, particularly for their application in Embodied AI. The benchmark is grounded in real-world injury narratives (NEISS) and industrial safety constraints. It consists of three components: ASIMOV-2.0-Injury (text-based risk assessment), ASIMOV-2.0-Constraints (image-based constraint adherence), and ASIMOV-2.0-Video (video-based intervention timing). The authors evaluate several state-of-the-art models, identify key "gaps" (modality, embodiment, latency), and propose a post-training paradigm to improve constraint adherence.

**Strengths:**

- Important and Timely Problem: The paper addresses the critical and increasingly relevant problem of physical safety for AI, a topic that is relatively under-explored compared to digital or content safety.

- Multi-Modal Scope: The effort to create a benchmark that spans multiple modalities (text, static images, and video) is a noteworthy step towards evaluating the complex perceptual-reasoning required for physical safety.

- Analysis of "Thinking" and Post-training: The paper provides an initial analysis of how inference-time compute ("thinking") and targeted post-training can impact safety adherence, which is a valuable area of investigation.

**Weaknesses:**

While the goal of ensuring physical safety for AI is critical, this paper suffers from several fundamental flaws in its premise, methodology, and positioning within the field.

**1. A Fundamental Misalignment with Embodied AI Principles**

- The paper's central claim is that it "comprehensively analyzes the ability of major foundation models to perceive risks, reason about safety, and trigger interventions" for Embodied AI systems. This claim is fundamentally **misleading**. True embodied intelligence is defined by its dynamic, closed-loop interaction with a physical environment. This involves a continuous cycle of **Perception -> Planning -> Action -> Feedback.** The agent's actions change the state of the world, and this change  leads to a new cycle.
- The proposed benchmark, however, evaluates none of this. The tasks in ASIMOV-2.0 are entirely disembodied and passive. They are, **in essence, static, multi-modal VQA or perception tasks:**
    * ASIMOV-2.0-Injury: A text-based question-answering task.
    * ASIMOV-2.0-Constraints: A visual question-answering (VQA) task that requires pointing to regions in a static image.
    * ASIMOV-2.0-Video: A video perception task that requires predicting a single timestamp.
- The paper merely assesses a model's ability to perceive potential risks in static media (text, images, videos). While perceptual safety is a component of embodied safety, it is not a sufficient proxy for it. A model might fail these perception-only tasks  but still perform safely in an embodied setup where it can actively query the environment. Conversely, a model that passes these static tests might catastrophically fail in a dynamic setting that requires real-time planning and adaptation.
- The paper fails to provide any evidence that the "safety risks" identified in this passive setting are the same ones that exist in an active embodied agent framework.

**2. Simplistic Task Design and Questionable Practical Utility**

This core flaw leads to the second major weakness: the tasks themselves are overly simplistic and their real-world utility is questionable.
* In ASIMOV-2.0-Constraints, the task is to point to objects that violate a constraint. This is a monolithic and artificial task format. A more meaningful evaluation would embed this constraint within a larger, multi-step embodied task (e.g., "Clean the room, but you cannot lift objects over 20kg"). This would test whether the agent's planning and action sequence adheres to the constraint, which is far more critical than a simple VQA pointing-task.
* In ASIMOV-2.0-Video, the task is to identify the "last possible timestamp" for intervention. It is entirely unclear what the real-world application of this is. A practical safety system does not ask for a post-hoc analysis of the "last possible" intervention point; it must proactively and continuously assess risk and intervene immediately when a threshold is crossed. This task format feels arbitrary and disconnected from any plausible engineering implementation of a safety intervention system.

**3. Critical Omission of Relevant Embodied AI Safety Benchmarks**

The authors' claim to be filling a gap in "physical safety for robots" is made without acknowledging the large and growing body of literature that does exactly this, but in actual embodied environments. The related work  conspicuously ignores numerous highly relevant benchmarks.

- The paper fails to cite, compare against, or differentiate itself from existing embodied AI safety benchmarks, such as SafeAgentBench，IS-Bench，EARBench，LabSafetyBench，Lota-Bench，SafePlan-Bench [1-5] and so on.
These benchmarks are explicitly designed to evaluate LLM-based agents in embodied environments. They directly test an agent's ability to follow safety rules, avoid hazards, not just passive perception.

- By ignoring this entire line of relevant work, the authors fail to (a) properly motivate their own benchmark, (b) demonstrate its novelty, and (c) justify why a disembodied, perception-only benchmark is a superior or even necessary contribution to a field that has already moved on to more complex, interactive evaluations.

**4. Over-reliance on Synthetic Data Generation**

While grounding in NEISS reports is a good starting point, the entire benchmark pipeline relies on generative models (LLMs, Imagen, Veo3) to create the scenarios and all testing media. This means the benchmark is not evaluating safety in the real world, but evaluating safety perception on synthetic media. The "long-tail" risks identified might be artifacts of the generative models themselves, rather than representative of physical reality. This adds another layer of abstraction that distances the evaluation from true "physical" safety.


**Reference:**

[1] SAFEAGENTBENCH: A BENCHMARK FOR SAFE TASK PLANNING OF EMBODIED LLM AGENTS

[2]Earbench: Towards evaluating physical risk awareness for task planning of foundation model-based embodied ai agents.

[3]Hazard challenge: Embodied decision making in dynamically changing environments.

[4]Is-bench: Evaluating interactive safety of vlm-driven embodied agents in daily household tasks.

[5]LabSafety Bench: Benchmarking LLMs on safety issues in scientific labs.

**Questions:**

None

---

> ### Author Response · Authors · 2025-11-20
> **We thank the reviewer for their feedback. We appreciate the opportunity to clarify the scope of our evaluation, the complexity of the tasks, and the rigorousness of our data generation pipeline.**
>
> We thank the reviewer for their feedback. We appreciate the opportunity to clarify the scope of our evaluation, the complexity of the tasks, and the rigorousness of our data generation pipeline.
> 1. **On the connection to Embodied AI (Response to "Fundamental Misalignment")**: We respectfully offer a different perspective on the reviewer’s concern that our work represents a misalignment with Embodied AI. We posit that evaluating the "brain" of the robot—specifically its ability to perceive risk and semantic severity across modalities—is a necessary (though not sufficient) condition for embodied safety. If frontier foundation models (Gemini, GPT, Claude) struggle to follow safety instructions or perceive risks in video/static contexts—as our results demonstrate—they cannot effectively function as safe "brains" in dynamic physical settings. We believe exposing these specific safety gaps before deployment is a critical contribution to the field.
> 2. **On the complexity of the Pointing Task**: Regarding the comment that the pointing task is "simplistic," we wish to highlight empirical evidence suggesting otherwise. While the task definition is straightforward, it remains highly challenging for current SOTA models. For context, benchmarks like PointArena (see https://pointarena.github.io/)  demonstrate a ~22% gap between human performance and state-of-the-art "embodied reasoning" models (including Gemini Robotics from Google DeepMind) on pointing tasks. Given that frontier models struggle significantly with this task in our benchmark, we argue it is a non-trivial and valuable metric for gauging fine-grained spatial reasoning gaps.
> 3. **On Proactivity and the "Last Intervention Timestamp"**: We fully agree with the reviewer’s statement that a safety system "must proactively and continuously assess risk." This is a core motivation of our work. We would like to direct the reviewer’s attention to Figure 10 (pp. 6–7), where we explicitly present a proactivity-vs-reactivity analysis. The "last intervention timestamp" is used precisely to measure this—penalizing models that fail to identify risk before a critical threshold. We believe this metric effectively captures the requirement for timely intervention without incurring the prohibitive costs of frame-by-frame annotation.
> 4. **Comparison with Related Benchmarks**: We thank the reviewer for suggesting these relevant works. We will cite them and discuss the following distinctions in the final version to clarify our contribution:
> - SafeAgentBench: Focuses on a very limited set of actions (e.g., "turn off," "pour"). Our benchmark provides comprehensive grounding in real-world injury narratives from US hospitals.
> - Earbench: Appears to be entirely synthetic without the human-annotated grounding or safety standard (NEISS/ISO) alignment present in our work. It also lacks injury severity metrics and video context.
> - HAZARD Challenge: Covers only three specific risk types (fire, flood, wind). Our work addresses the "long tail" of diverse safety risks grounded in real-world data.
> - Is-Bench & Lab Safety Bench: These benchmarks do not cover the physical constraint adherence or video modalities central to our study, nor do they cover the diversity of NEISS-grounded tail risks.
> 5. **On Data Quality and "Over-reliance on Synthetic Data"** We wish to clarify our data generation pipeline to address concerns about synthetic quality. While we utilize synthetic generation for scalability, human verification is central to our methodology. As detailed in the paper, every evaluation instance underwent multiple rounds of human annotation to check for:
> - Photorealism.
> - Realistic physics.
> - Object permanence and consistency.
> - Violations of cause-and-effect logic.
> - Any other unrealistic artifacts.
>
> We aggressively filtered out instances that lacked strong human consensus. Consequently, only a small fraction of the generated data met the strict quality standards required for inclusion in the final benchmark. We believe this hybrid approach ensures the data is both scalable and rigorously grounded.

---

> > ### Comment · Reviewer_oYkf · 2025-11-25
> >
> > Thank you for the author's detailed reply. The author resolved some of my issues, so I intend to raise my score.  I would like to see a revised manuscript.

---

> ### Author Response · Authors · 2025-11-25
> **Related works section updated**
>
> Dear reviewer oYkf,
>
> We have made a first pass at updating the related works section and we have included references to the papers and prior work which you have shared with us.
>
>
> Thanks a lot for sharing this useful prior work and helping us improve the paper.

---

### Official Review · Reviewer_6BXZ · 2025-10-31

**Soundness:** 3
**Presentation:** 3
**Contribution:** 3
**Rating:** 6
**Confidence:** 3

**Summary:**

The paper presents a multimodal benchmark for physical safety across text, image, and video. The work systematically evaluates frontier models on latent risk identification, action outcome assessment, embodied constraint adherence, and the timing of video-based interventions. It further investigates how increased “thinking” budgets (chain-of-thought/reasoning compute) and post-training (SFT+RL) affect adherence to safety constraints, showing that small safety datasets can substantially reduce violation rates.

**Strengths:**

1、The benchmark Grounding in NEISS narratives and industrial safety standards strengthens real-world alignment and supports scalability.

2、The generate–critic–refine pipeline improves data quality and enables continual dataset evolution.

3、The paper is easy to follow.

**Weaknesses:**

1、The heavy reliance on Imagen/VEO3-generated images and videos may limit external validity due to realism gaps and distribution shift. I recommend adding a small real-image/video validation slice and reporting cross-domain consistency, including absolute performance deltas and rank stability between synthetic and real subsets.

2、The Action outcomes, severity, and intervention timing depending on contextual constraints (reachability, latency) may lead to the disagreement. Aggregating with 60–80% consensus thresholds discards informative ambiguity. I suggest preserving multi-annotator label distributions and evaluating with soft labels or proper scoring rules to test whether models show low confidence or cautious behavior on high-disagreement samples.

3、The evaluation metrics lack sufficient interpretability.The constraint violation metric penalizes any point falling into a violating box, but task difficulty varies with constraint type, scene complexity, and occlusion. I think you could provide stratified curves by scene complexity (object count, occlusion, scale variation) and by the number of composed constraints to localize capability bottlenecks more precisely.

**Questions:**

See Weakness

---

> ### Author Response · Authors · 2025-11-20
> **Thank you for the high quality feedback and suggestions for further refining the benchmark.**
>
> We thank the reviewer for their detailed and high-quality feedback. We are encouraged that the reviewer found the work "easy to follow" and recognized the value of our "Grounding in NEISS narratives" and the "generate–critic–refine pipeline". We agree that addressing the sim-to-real gap and preserving ambiguity are critical next steps. We address the specific recommendations below.
> 1. **Response to "Realism Gap" & Real-World Validation**: The reviewer suggests adding a "real-image/video validation slice" to measure cross-domain consistency.
> To address these concerns, we asked human annotators to rate on photorealism of the synthetically generated videos.  As part of our future work, we are working on releasing an evaluation set with videos at ALOHA performing some of the tasks in the ASIMOV-2 Constraints.
> It is extremely hard (if not impossible) to build a real dataset for ASIMOV-2 Injuries and ASIMOV-2 Videos (At least the way the tasks are currently formulated) as they both feature potential injuries to human beings. To the best of our knowledge, we haven’t come across any real image/video datasets relevant for robotics featuring injury prevention.
>
> 2. **Response to "Ambiguity & Soft Labels"**:  The reviewer notes that thresholding consensus (we used 60-80% ) discards informative ambiguity regarding uncertainty.
> We are happy to confirm that the benchmark is fully open. Since the review process is anonymous, we will include the website link in the final version of this paper. The constraints component is released with a full set of labels, so it’s easier for researchers to compute soft label based metrics as per their requirements. We will also commit to releasing full label distribution for all benchmarks.
> 3. **Response to "Stratified Metrics & Complexity"**:
> This is great feedback, we don’t have human annotations for occlusion, scene complexity currently. We will try to cover some of these before the final version of the paper. Our earlier results showed that having too many objects in the image (over 6) resulted in a sharp decline in performance. Given that all the current SOTA models seem to struggle with this task, we dropped such instances from the benchmark.

---

### Official Review · Reviewer_gjnM · 2025-11-01

**Soundness:** 3
**Presentation:** 3
**Contribution:** 2
**Rating:** 4
**Confidence:** 3

**Summary:**

This paper introduces ASIMOV-2.0, a comprehensive benchmark suite designed to evaluate whether AI systems—especially multimodal and embodied ones—can perceive physical danger and intervene appropriately. The benchmark spans three modalities: text (ASIMOV-Injury), image (ASIMOV-Constraints), and video (ASIMOV-Video), grounded in real-world hospital injury narratives and operational safety standards. The authors assess leading proprietary models (GPT-5, Claude Opus 4.1, Gemini 2.5 Pro) and reveal three consistent gaps: a modality gap, an embodiment gap, and a latency gap (smaller, faster models perform worse). Finally, they propose a “thinking-for-safety” paradigm, where explicit reasoning traces and post-training improve safety constraint adherence and interpretability.

**Strengths:**

- This paper aims to address an underexplored yet crucial domain, physical safety in embodied AI, through a multimodal, real-world-grounded benchmark.

- The authors evaluate several state-of-the-art models and uncovers distinct modality, embodiment, and inference-budget gaps. They demonstrate that structured "thinking" traces and post-training can make safety reasoning more transparent and consistent across tasks.

**Weaknesses:**

- The use of large models as critics in the data-generation loop raises concerns—without independent human auditing, bias and hallucination could propagate, undermining benchmark validity.

- Although grounded in injury data, the benchmark’s images and videos are AI-generated, and may not capture real-world physics or social context faithfully; this limits external validity.

- Evaluation excludes open-source and mid-tier models, restricting reproducibility and the generality of conclusions across the model landscape.

- While post-training improves safety scores, the paper does not quantify potential degradation in general reasoning or task performance, raising the risk that safety gains may come at the expense of versatility.

**Questions:**

- How is the critic model’s reliability validated during the generator–critic–refine loop? Are there calibration or cross-checking mechanisms with human reviewers?

- How realistic are the AI-generated visual scenes? Have the authors measured perceptual realism (e.g., via human evaluation or physics consistency metrics)?

- Why were open-source foundation models (e.g., LLaVA, Qwen-VL, or Kosmos series) excluded? Would the benchmark be publicly released to support reproducibility?

- How does “thinking-for-safety” affect broader capabilities—e.g., general reasoning, speed, or creativity? Is there evidence that the approach scales without diminishing non-safety performance?

---

> ### Author Response · Authors · 2025-11-20
> **Thanks a lot for the insightful and constructive review. We have tried to address your concerns about "Safety tax" in a new section in the appendix.**
>
> We thank the reviewer for their constructive feedback and for recognizing the importance of benchmarking physical safety in embodied AI. We appreciate the specific questions regarding data validation and model coverage, which we address below.
> 1. **Response to Benchmark Validity & "Critic" Reliability**: The reviewer raises a valid concern about potential bias propagation if the "Critic" model acts as the sole filter.
> - *Human-in-the-Loop Verification*: We clarify that the "Critic" model (LLM) served only as an initial automated filter. The final validity of the benchmark was established via rigorous human annotation. As detailed in Section 2, we obtained "ground-truth” answers with 5 human raters per instance" for all three of our benchmarks.
> - *Quality Control*: We explicitly filtered out the data where raters had low consensus due to ambiguity". For video, we "selected only those videos where intervention timestamps provided by the human raters had a standard deviation below 1.0s". This ensures that any "hallucinations" passed by the Critic were caught and removed by human annotators.
> 2. **Response to "Realism" of Synthetic Scenes**: The reviewer questions if synthetic scenes capture real-world physics.
> - *NEISS Grounding*: To ensure the scenarios are realistic (even if pixels are synthetic), we grounded our generation in the "National Electronic Injury Surveillance System (NEISS)" , which provides "narrative descriptions of real injuries to humans ” providing a rich sampling of the 'long-tail' of physical safety risks". Asimov 2 -Injury and Asimov  2- Video instances/scenarios are both therefore grounded in NEISS. Asimov 2 - Constraints taxonomy of harms is grounded on the ISO standards and focuses on adherence to very practical and real constraints such as payload limits.
> - *Ensuring Visual and Physics Realism*: For realism, human raters specifically "filtered out videos which had quality issues like lack of photorealism, violation of laws of physics etc.". Multiple human annotators were provided examples of such violations to ensure high data quality.
>
> 3. **Response to Open Source & Reproducibility**: The reviewer asks about the exclusion of open-source models and benchmark availability.
> - *Benchmark Availability*: We are happy to confirm that the benchmark is fully open. Since the review process is anonymous, we will include the website link in the final version of this paper.
> - *Open Models*: In this paper, we focused on safety evaluation of ALL major frontier models used billions of times per day as a priority. Our analysis demonstrates the performance gap between text/video injury intervention, gaps between large/small models and gaps in adhering to constraints. We will add an additional section on Gemma models in response to the reviewers suggestion; however we expect our insights and conclusions to remain the same.
>
> 4. **Response to "Safety Tax" on General Capabilities**:
> This is a great consideration and to address this concern we have added a section in the Appendix which shows there is little or no significant decline in underlying pointing ability of the model. We compared baseline and finetuned models on the POINT-BENCH, and we see a very negligible (but not statistically significant) impact on pointing average accuracy.

---

> > ### Author Response · Authors · 2025-11-25
> >
> > Dear Reviewer gjnM,
> >
> > We hope you had a good weekend.
> >
> > We wanted to briefly follow up on the rebuttal we posted late last week. We have done our best to address your main concerns, specifically regarding the safety alignment tax.
> >
> > We would greatly value your thoughts on our response while the discussion period is still active. If our revisions have resolved your concerns, we would appreciate it if you would consider updating your evaluation.
> >
> > Thank you again for your time and help in improving our paper.

---

### Official Review · Reviewer_JKqL · 2025-11-09

**Soundness:** 3
**Presentation:** 4
**Contribution:** 3
**Rating:** 6
**Confidence:** 3

**Summary:**

The paper introduces a benchmark that evaluates whether AI models can perceive physical harm. The benchmark spans text, image and instructions, and video modalities, allowing comprehensive assessment across different forms of physical-risk understanding. The authors analyze how popular models perform on identifying risks over time, and propose a post-training paradigm that teaches models to reason about physical safety constraints, which improves physical safety performance.

**Strengths:**

1. The problem is practical and important, especially for real-world embodied AI systems. As far as I know, this is the first benchmark addressing physical safety across text, image, and video modalities.

2. The dataset is grounded in reliable and authoritative sources. For example, the authors use the NEISS system, which collects injury reports from approximately 100 hospitals in U.S.. They also incorporate industrial safety standards such as ISO 10218-1:2025 and ISO/TS 15066:2016.

3. The benchmark addresses operational safety, which is particularly significant for embodied AI systems.

4. The authors evaluate several popular models on their abilities with the benchmark.

5. The writing is clear and easy to understand.

**Weaknesses:**

1. The generated images and videos might fail to relect real-world cases, especially for task-specific embodied AI systems.

2. The data size is small, which might fail to capture edge cases in real-world scenarios.

3. Embodied AI systems can handle complex tasks such as perception, reasoning, and physical actions, and therefore rely on different hardware such as cameras, and different sensors. Therefore, there may be a gap between the data an embodied system collects and the data used in the benchmark.

**Questions:**

1. There may be a gap between the data an embodied system collects and the data used in the benchmark. A real embodied agent may leverage sensors to detect signals or subtle movements, instead of images or videos. However, such data are not explicitly represented in the benchmark. Could you provide more explanation on how the benchmark could be extended to match real-world use cases?

2. The benchmark identifies physical safety issues. But how these evaluations relate to concrete tasks performed by real-world embodied agents? Could you provide some practical use cases?

---

> ### Author Response · Authors · 2025-11-20
> **Thank you for your insightful review, we have tried to address points raised by you and shared our perspective on your concerns. Happy to discuss further to align further in our shared understanding.**
>
> We thank the reviewer for their detailed assessment  and for recognizing the novelty of ASIMOV-2.0 in addressing physical safety across modalities. We particularly value the reviewer's insight regarding the "Sensor Gap" in embodied systems. We address the specific concerns regarding real-world sensors and concrete use cases below.
>
>  We prioritized RGB data because the current frontier of VLA models (e.g., Gemini Robotics) relies heavily on visual semantic reasoning to interpret complex scene dynamics. ASIMOV-2.0 serves as a comprehensive evaluation for this visual safety backbone.
> However, our benchmark is designed to be extensible. As noted in our taxonomy (Fig. 12), we cover constraints like "Payload" and "Thermal". While currently inferred visually (e.g., seeing steam), the benchmark can be extended to match real-world use cases by:
> - **Multimodal Prompts**: Injecting sensor logs alongside the image to test if the model can cross-reference visual data with tactile limits.
> - **Depth Channels**: Future iterations can include depth maps as additional image channels to test geometric safety constraints more directly.
>
> 2. **Concrete Use Cases & Task Relation (Answering Q2)** The reviewer asks how ASIMOV evaluations map to concrete tasks. A failure on our metrics serves as a predictor for specific real-world failure modes:
> - *Use Case A (Industrial Cobot - ASIMOV-Video)*: Consider a collaborative robot working alongside humans. Our "Last Intervention Timestamp" metric directly maps to the robot's emergency stop latency. If a model performs poorly here (as we observed with smaller models), the agent will fail to halt its trajectory when a human worker enters its workspace, leading to collision.
> - *Use Case B (Domestic Service - ASIMOV-Constraints)*: Consider a robot tasked with clearing a table. Our "Gripper Geometry" constraint task evaluates if the model understands its own physical limits. A failure here means the robot might attempt to grasp a heavy object (e.g., a cast-iron skillet) that exceeds its payload or grasp a sharp knife by the blade, resulting in equipment damage.
> 3. **Response to "Synthetic Data & Edge Cases"**: We acknowledge the concern regarding the  data size.
> - *Safety Density vs. Volume*: While smaller than general pre-training datasets, ASIMOV-2.0 focuses on the "long tail" of low-likelihood, high-risk scenarios. Standard robot datasets contain millions of frames of successful tasks but almost no accidents. By seeding our data with NEISS injury reports (e.g., specific machinery fractures, burns), we capture edge cases that are absent from real-world training data but critical for safety validation.
>
> In this work we focused on getting high quality human annotations for our dataset, so it’s well grounded, and doesn’t have low quality  artifacts from the synthetic generation process. Due to budget constraints associated with high quality human annotations, we were operating with a practical limit on the size of the benchmark.

---

### Note · Program_Chairs · 2026-01-17
**Submission Desk Rejected by Program Chairs**

The following references in this submission do not refer to real documents and/or have major errors in bibliographic information:

 Chien-Yi Wu et al. Project gazelle: A multimodal ai model for meta's next-generation smart glasses. Meta AI Research Blog, October 2024. URL https://ai.meta.com/blog/ project-gazelle-meta-next-generation-smart-glasses-ai-model